# Design and Validation of the Trailing Edge of a Variable Camber Wing Based on a Two-Dimensional Airfoil

**DOI:** 10.3390/biomimetics9060312

**Published:** 2024-05-23

**Authors:** Jin Zhou, Xiasheng Sun, Qixing Sun, Jingfeng Xue, Kunling Song, Yao Li, Lijun Dong

**Affiliations:** 1Chinese Aeronautical Establishment, Beijing 100012, China; sunxs623@aliyun.com (X.S.); sunqx@avic.com (Q.S.); xuejingfeng@cae.ac.cn (J.X.); songkunling@cae.ac.cn (K.S.); donglj001@avic.com (L.D.); 2School of Aeronautics and Astronautics, Shanghai Jiao Tong University, Shanghai 200240, China; 3Aircraft Strength Research Institute of China, Xi’an 710065, China; liyao0313@sina.com

**Keywords:** variable camber wing, trailing edge, Watt I mechanism, full-size test pieces, ground validation

## Abstract

Variable camber wing technology stands out as the most promising morphing technology currently available in green aviation. Despite the ongoing advancements in smart materials and compliant structures, they still fall short in terms of driving force, power, and speed, rendering mechanical structures based on kinematics the preferred choice for large long-range civilian aircraft. In line with this principle, this paper introduces a linkage-based variable camber trailing edge design approach. Covering coordinated design, internal skeleton design, flexible skin design, and drive structure design, the method leverages a two-dimensional supercritical airfoil to craft a seamless, continuous two-dimensional wing full-size variable camber trailing edge structure, boasting a 2.7 m span and 4.3 m chord. Given the significant changes in aerodynamic load direction, ground tests under cruise load utilize a tracking-loading system based on tape and lever. Results indicate that the designed single-degree-of-freedom Watt I mechanism and Stephenson III drive mechanism adeptly accommodate the slender trailing edge of the supercritical airfoil. Under a maximum cruise vertical aerodynamic load of 17,072 N, the structure meets strength requirements when deflected to 5°. The research in this paper can provide some insights into the engineering design of variable camber wings.

## 1. Introduction

Commercial airliners contribute significantly to carbon emissions within the aviation industry. The albatross can soar for hundreds of kilometers continuously without flapping its wings. It locks its joints while cruising and then adjusts its wing feathers to keep it in the best cruising condition at all times. Inspired by this, for long-range commercial aircraft, it is possible to design a smooth, continuous, seamless wing that allows the aircraft to maintain optimal cruising efficiency without having to adjust its flight altitude at all times. Variable camber wing technology offers substantial potential for enhancing the performance of large long-range civil aircraft through dynamic shape adaptation [1]. During cruising, adjusting the camber of the variable camber trailing edge (VCTE) minimizes resistance and enhances the lift-to-drag ratio. Additionally, the absence of seams and hinges in the VCTE ensures smooth airflow transitions, thereby reducing noise during takeoff and landing operations effectively [2,3].

The development of variable camber wings can be delineated into three distinct stages. The initial stage (1970s–1980s), primarily centered on the structural design of conventional rigid mechanisms, aimed at verifying the feasibility of variable camber technology concerning aerodynamics, noise reduction, and control. However, this era encountered numerous challenges, including excessive structural weight and low reliability. A notable project from this period was the Mission Adaptive Wing (MAW) project in the United States [4]. In the subsequent stage (1990s–2000s), emphasis shifted toward the utilization of smart materials such as piezoelectric materials and shape memory alloys to achieve lightweight structural design and actuation. A prominent project during this phase was the Smart Wing project in the United States. The third stage (2010s–2020s), primarily from an engineering standpoint, underscored the necessity of integrating traditional rigid mechanisms, novel sensors, and multifunctional materials to craft ingenious structures that meet the demands for intelligence and lightweight properties. Noteworthy projects in this era include the Smart Intelligent Aircraft Structures (SARISTU) in Europe [5] and the Adaptive Compliant Trailing Edge (ACTE) in the United States [6,7]. This evolution demonstrates a progression from the singular focus on functional realization to the pursuit of lightweight structures and eventually to the consideration of the balance between function and structure for engineering applications, with a corresponding continuous enhancement in the Technology Readiness Level (TRL). According to the “Aircraft Technology Roadmap to 2050” by IATA [8], retrofitting variable camber wing technology before 2030 could yield fuel reduction benefits ranging from 1% to 2%, while incorporating variable camber concepts with new control surfaces could potentially achieve fuel reduction benefits of 5% to 10%. Furthermore, the Aerospace Technology Institute (ATI) of the UK released the “Zero Emission Flight” (Fly Zero) aerodynamic structures technology roadmap in March 2022, identifying morphing trailing edges and wing span morphing as among the most promising technologies, with the potential to achieve TRL6 status before 2030.

Numerous studies presently concentrate on employing smart materials for VCTE design, particularly targeting small aircraft or Unmanned Aerial Vehicles (UAVs) [9,10,11,12,13]. This is mainly due to the limited design space of small aircraft, relatively lower flight speeds, and deflection angles. Smart materials such as shape-memory alloys, shape-memory polymers, and piezoelectric composites are better suited to accommodate these limitations, facilitating more compliant deformation. Conversely, for larger aircraft, the influence of aerodynamic loads and greater deflection requirements necessitates the utilization of rigid mechanisms to reconcile the trade-off between high load-carrying capacity and substantial deformation. Hence, guided by engineering principles, this paper investigates VCTE design methodologies encompassing deformation skeletons, flexible skins, and structural configurations, among others, especially the mechanism design. Subsequently, leveraging the two-dimensional airfoil of the Chinese Aeronautical Establishment—Aerodynamic Validation Model (CAE-AVM) [14,15], it was stretched 2.7 m along the span direction, and a full-scale VCTE was meticulously designed and subjected to ground testing. According to our other research evaluation, employing the VCTE can enhance the lift-to-drag ratio of the aircraft airfoil by 7% and the overall lift-to-drag ratio by 3.5%, and, in total, save 8760 tons of fuel during the service life per aircraft [16].

## 2. Design Concept

### 2.1. Design Concept and Implementation Approaches

In designing the VCTE, it is imperative to concurrently fulfill three fundamental requirements: smooth deformation, aerodynamic load-carrying capability, and structural lightweightness, as shown in Figure 1. Smooth deformation means that the deformation angle and rate of the structure align with aerodynamic specifications, resulting in a smooth, wrinkle-free shape post-transformation; aerodynamic load-bearing capability necessitates the structure’s ability to withstand high-speed aerodynamic loads and internal deformation-driving loads without succumbing to excessive elastic deformation; and structural lightweightness mandates that the variable camber implementation should not incur excessive structural costs, ensuring good reliability and maintainability.

Conventional VCTE, for example, last-century designs, meets deformation and aerodynamic load-bearing requirements but suffers from weightiness, complexity, and high costs. Flexible structures, while lightweight and capable of substantial deformation, are limited in the loads they can bear, making them suitable primarily for low-speed scenarios. Lightweight topological structures [17,18,19], such as lattice-based designs [20], face challenges in achieving large deformations and require heightened attention to structural load-carrying capacity and reliability concerns. Ongoing research and projects concerning VCTE are progressively addressing these three aspects, striving to attain a harmonious balance among them.

Following decades of research on VCTE, various implementation approaches have emerged, as illustrated in Figure 1. Conventional structures exhibit drawbacks such as excessive structural weight and low reliability. Improved variable camber structures, relying on lift augmentation devices, have found application in aircraft like the B787 and A340/A350; however, they face challenges related to insufficiently smooth shape transitions. Smart materials, such as shape memory materials and piezoelectric materials [21,22], offer flexibility and significant potential, yet simultaneously addressing driving force, speed, and stroke remains challenging. A comparison of these three implementation approaches is detailed in Table 1.

### 2.2. Discussion of Structure-Based Design and Aerodynamic-Based Design

In the design of a wing, it is customary to initially establish the aerodynamic profile before configuring the internal structure of the wing according to aerodynamic requirements. However, with variable camber wings capable of adjusting to various flight conditions through continuous deflection, multiple shapes must be concurrently considered during the design process. Theoretically, the optimal approach involves directly deriving a series of shapes through aerodynamic design, with the structure subsequently endeavoring to realize these target shapes.

However, the current main challenge in variable camber wing technology lies in structural realization and verification, which remains relatively underdeveloped. Designing a structure solely based on a sequence of aerodynamic shapes often proves challenging, as it must simultaneously fulfill requirements for mechanism movement, aerodynamic load-bearing capacity, and lightweight construction. Therefore, the fundamental concept of this study is as follows: Firstly, instead of determining a series of initial shapes, an initial shape is established. Subsequently, the required deformation angle is determined based on aerodynamic analysis results. Following this, considering the interplay between aerodynamics and structure, the deformation mechanism is designed to ultimately ascertain a sequence of shapes. Thereafter, aerodynamic analysis of these shapes is conducted, followed by iterative design iterations. In contrast to conventional wings, the design of variable camber wings necessitates a greater emphasis on mechanism design, given that the weight and stability of the mechanism pose the primary challenge and limitation.

## 3. Mechanism Design Method of VCTE Based on Planar Linkage

To meet the variable camber requirements, the structure must be partitioned into distinct components to collaboratively fulfill their respective functions. The aerodynamic load is sustained by the integrated deformation skeleton mechanism, while the internal driving load is supported by the drive system. Additionally, the airfoil adjustments are facilitated by a flexible skin to ensure seamless and coordinated contours during the deformation process, and ideal skin is characterized by low in-plane stiffness and high out-of-plane stiffness. Initially, the movement pattern of the deformation skeleton mechanism, the driving mechanism of the drive system, and the characteristics of the flexible skin are determined. As the deformable skeleton acts as the intermediary structure between the drive system and the flexible skin, it dictates the form of the drive and the structural configuration of the flexible skin. Subsequently, mechanism optimization is conducted based on the movement points of the deformed skeleton, leading to further refinement of the structural details. Finally, ground tests are conducted through the manufacturing of test specimens.

Planar linkages are the most commonly utilized mechanisms, offering advantages such as the ability to transmit substantial loads, adaptability to complex motion curves, and precision in manufacturing and assembly. In this paper, the concept of a “finger” is adopted, dividing VCTE into multiple rigid blocks to simulate the desired shape during the deformation process. The following issues require clarification:Ensuring coordinated deformation between the skin and skeleton during the deflection process.Managing the relative movement of connecting rods and rigid body blocks throughout the skeleton’s deformation process.Determining the form and parameters of the driving skeleton.

The coordination of skin and skeleton deformation establishes the general design direction of the mechanism, while the design of rigid body blocks influences the configuration of the connecting rods. Additionally, the design of the drive system is contingent upon the movement space of the mechanism, as well as internal and external loads.

### 3.1. Coordinated Design of Skin and Skeleton

The VCTE consists of upper and lower skins, with bending-induced camber changes inevitably affecting their relative lengths. Various skin length variation schemes can be designed according to different concepts. The upper skin can adopt three states: stretched, unchanged, and compressed, mirroring the options available for the lower skin, resulting in N = 3 × 3 = 9 states in total. Excluding the four illogical states where the upper skin remains unchanged while the lower stretches, or the upper skin shrinks while the lower remains unchanged, or the upper skin shrinks while the lower stretches, or both skins are compressed (reducing aerodynamic performance), five feasible design ideas are proposed, illustrated in Figure 2:Mode 1: The length of the upper skin remains unchanged while the lower wing skin shrinks;Mode 2: The length of the lower skin remains unchanged, and the upper wing skin is stretched;Mode 3: The upper wing skin stretches while the lower skin shrinks (typically, the length of the camber line remains unchanged);Mode 4: Both the upper wing skin and the lower skin stretch simultaneously;Mode 5: The lengths of the upper and lower skins remain unchanged, with slipping and displacement occurring at the trailing edge tip [23].

In designing the skin, two approaches are viable: an overlapping design, where two sections slide relative to each other on the overlapping surface, or a flexible skin design, incorporating an elastic material in the connection area to enable stretching and shrinking. Prioritizing aerodynamic efficiency, skin stretching is favored over shrinkage whenever feasible, as it tends to increase wing area and subsequently enhance lift.

Regarding the complexity of skin adjustment, Modes 1 and 2 are relatively straightforward, involving changes to only a portion of the skin (upper or lower); Modes 3 and 4 are more intricate, requiring alterations to both upper and lower skins; whereas Mode 5 is the simplest, requiring no adjustment to skin length. Regarding the complexity of the skeleton, Modes 1 and 2 pose challenges as the rotation center must be positioned on the side of the skin and the other side maintains a constant length during deformation. Mode 3 is comparatively straightforward to implement, as the rotation center can be positioned on or near the camber line. Mode 4 involves both rotation and translation in the skeleton, allowing for flexible design possibilities. Mode 5 necessitates a complex design of the sliding connection between the skeleton and the skin. Based on these design considerations, various VCTE structures can be explored. This study adopts Mode 3, where the VCTE is divided into multiple rigid body blocks. Each block rotates relative to the camber line, enabling the upper skin to stretch and the lower skin to shrink. This approach offers a relatively simple design concept, with a straightforward skeleton design. Utilizing the camber line as the target for evaluating different deflection shapes facilitates the optimization of the structure to approximate various deflection shapes effectively.

### 3.2. Selection of Linkage Mechanism Form

Common planar linkage mechanisms include four-link, six-link, eight-link, etc. The complexity of these mechanisms gradually increases, but the realized motion curves can also be more complex.

The four-bar linkage, a single-Degree-of-Freedom (DOF) mechanism, consists of four connecting rods connected by four rotating nodes. However, it cannot simultaneously drive three rigid blocks. In contrast, the six-bar linkage, also a single-DOF mechanism, connects six links through seven nodes. Two topological configurations exist for the links in a six-bar linkage: the Watt chain and the Stephenson chain. The Watt chain comprises two loops, divided into two types: Watt I and Watt II. The Stephenson chain consists of one loop with four bars and one loop with five bars, categorized into three types: Stephenson Type I, Type II, and Type III. Figure 3 illustrates the specific configurations of the Watt chain and Stephenson chain [24].

The eight-bar linkage yields more complex motion and enables the attainment of more ideal shapes, it also introduces increased friction, structural weight, and manufacturing complexity. Consequently, the utilization of a planar six-bar linkage mechanism in VCTE design emerges as the outcome of comprehensive optimization across multiple factors.

### 3.3. Designing Three Blocks Using the Watt Mechanism

Based on the trade-off between aerodynamic efficiency and complexity, the Watt mechanism is chosen to actuate the trailing edge structure. Given that Watt Type I offers a more compact design compared to Watt Type II, it is preferred. First, three connected continuous deflection members corresponding to the blocks in the topological configuration were identified. Since adjacent rigid body blocks require relative rotation angles, the selection of the bar path forming the deflection blocks should not involve choosing two edges of the triangle simultaneously. As illustrated in Figure 4, three optional paths from the starting fixed points are A -> B -> C -> G, A -> E -> G -> C, and D -> F -> C -> G.

When the VCTE deflects downward, the rear block of the three blocks deflects downward sequentially relative to the front block. Therefore, during path design, it is crucial to ensure consistency in the relative rotation directions of the connected blocks, as depicted on the right side in Figure 5.

In the preliminary point design phase, the method involves initially fixing three connecting blocks in sequence according to the airfoil shape as illustrated in Figure 4. Subsequently, the positions of other nodes in the Watt I-type six-bar linkage are determined based on the “connecting block intersection” method described in Figure 5. When the VCTE deflects downward, it is essential to ensure that the deflecting direction of the middle block relative to the front block and the rear block relative to the middle block is consistent. Consequently, lines AB and DF must intersect to ensure uniform deflection of AB and BC in the same direction, as depicted on the right side of Figure 5. Conversely, on the left side of Figure 5, the deflecting direction of AB and BC are opposite. According to this method, four primary mechanism variations can be derived, as depicted in Figure 6.

Of the four mechanism forms, the first and third are symmetrical and suitable for narrower and longer structures, whereas the second and fourth are structurally similar and suitable for more compact configurations, or for symmetrical airfoil configurations, such as diamond-shaped airfoils. Supercritical airfoils, commonly employed in large commercial aircraft, feature thinner trailing edges owing to the after-loading effect. Therefore, the first form is preferable for arranging linkages to avoid interference. In order to display the three motion rigid blocks formed more intuitively, a virtual H point is introduced on the line CG, and the line CG is expanded into Lump-CGH. When designing according to the trailing edge airfoil, the H point can be set as the tip of the trailing edge. The resulting mechanism is illustrated in Figure 7.

### 3.4. Kinematics Analysis

To streamline the deformation design process, the vector calculation method is employed, illustrated in Figure 8. The X1AY1 coordinate system is defined with hinge A as the origin and the line connecting hinges D and A as the Y1 axis. Paths A -> B -> F -> D and A -> E -> G -> C -> F -> D are selected to form a closed vector position equation system for the mechanism.
(1)$$\left\{\begin{array}{c} l_{1}e^{\theta_{0}i}+l_{2}e^{\theta_{1}i}-l_{3}e^{\theta_{2}i}-l_{4}e^{-\frac{\pi }{2}i}=0\\ l_{5}e^{\left(\theta_{0}-∠BAE\right)i}+l_{6}e^{\theta_{3}i}-l_{7}e^{\theta_{4}i}-l_{8}e^{\theta_{5}i}-l_{3}e^{\theta_{2}i}-l_{4}e^{-\frac{\pi }{2}i}=0 \end{array}\right.$$
where $\theta_{5}=∠BFC-\left(\pi -\theta_{1}\right)=∠BFC+\theta_{1}-\pi$.

Expand the real and imaginary parts of the equation,
(2)$$\left\{\begin{array}{c} \begin{array}{c} l_{1}\cos\theta_{0}+l_{2}\cos\theta_{1}-l_{3}\cos\theta_{2}-l_{4}\cos\left(-\frac{\pi }{2}\right)=0\\ l_{1}\sin\theta_{0}+l_{2}\sin\theta_{1}-l_{3}\sin\theta_{2}-l_{4}\sin\left(-\frac{\pi }{2}\right)=0 \end{array}\\ \begin{array}{c} l_{5}\cos\left(\theta_{0}-∠BAE\right)+l_{6}\cos\theta_{3}-l_{7}\cos\theta_{4}-l_{8}\cos\left(∠BFC+\theta_{1}-\pi \right)-l_{3}\cos\theta_{2}-l_{4}\cos\left(-\frac{\pi }{2}\right)=0\\ l_{5}\sin\left(\theta_{0}-∠BAE\right)+l_{6}\sin\theta_{3}-l_{7}\sin\theta_{4}-l_{8}\sin\left(∠BFC+\theta_{1}-\pi \right)-l_{3}\sin\theta_{2}-l_{4}\sin\left(-\frac{\pi }{2}\right)=0 \end{array} \end{array}\right.$$

Since $\theta_{5}$ is dependent on $\theta_{1}$, there are only four unknowns, corresponding to the four equations available, enabling us to determine the position. The following formula is derived using trigonometric functions:(3)$$\left\{\begin{array}{c} \begin{array}{c} l_{2}\cos\theta_{1}-l_{3}\cos\theta_{2}+l_{1}\cos\theta_{0}=0\\ l_{2}\sin\theta_{1}-l_{3}\sin\theta_{2}+{l_{1}\sin\theta_{0}+l}_{4}=0 \end{array}\\ \begin{array}{c} l_{8}\cos∠BFC\cos\theta_{1}-l_{8}\sin∠BFC\sin\theta_{1}-l_{3}\cos\theta_{2}+l_{6}\cos\theta_{3}-l_{7}\cos\theta_{4}+l_{5}\cos\left(\theta_{0}-∠BAE\right)=0\\ -l_{8}\sin∠BFC\cos\theta_{1}-l_{8}\cos∠BFC\sin\theta_{1}-l_{3}\sin\theta_{2}+l_{6}\sin\theta_{3}-l_{7}\sin\theta_{4}+l_{5}\sin\left(\theta_{0}-∠BAE\right)-l_{4}=0 \end{array} \end{array}\right.$$

Let $\cos\theta_{i}=x_{i}$, then $\sin\theta_{i}=\pm \sqrt{1-x_{i}^{2}}$, employ Newton’s iteration formula $x^{\left(k+1\right)}=x^{\left(k\right)}-{\left[F^{\prime }\left(x^{\left(k\right)}\right)\right]}^{-1}F\left(x^{\left(k\right)}\right)$:(4)$$F\left(x\right)=\left\{\begin{array}{c} \begin{array}{c} l_{2}x_{1}-l_{3}x_{2}+l_{1}\cos\theta_{0}\\ l_{2}\sqrt{1-x_{1}^{2}}-l_{3}\sqrt{1-x_{2}^{2}}+{l_{1}\sin\theta_{0}+l}_{4} \end{array}\\ \begin{array}{c} l_{8}\cos∠BFCx_{1}-l_{8}\sin∠BFC\sqrt{1-x_{1}^{2}}-l_{3}x_{2}+l_{6}x_{3}-l_{7}x_{4}+l_{5}\cos\left(\theta_{0}-∠BAE\right)\\ -l_{8}\sin∠BFCx_{1}-l_{8}\cos∠BFC\sqrt{1-x_{1}^{2}}-l_{3}\sqrt{1-x_{2}^{2}}+l_{6}\sqrt{1-x_{3}^{2}}-l_{7}\sin\sqrt{1-x_{4}^{2}}+l_{5}\sin\left(\theta_{0}-∠BAE\right)-l_{4} \end{array} \end{array}\right.$$

### 3.5. Drive Mechanism

The drive mechanism should be installed within the trailing edge as much as possible to avoid interference with the wing fuel tank within the wing. When selecting the driving method, it is crucial to consider both the driving stroke and the layout space. If the driving mechanism is placed on the Lump-ABE, it results in limited structural design space and potential interference issues. Conversely, positioning the driving mechanism on the block Lump-CGH increases the structural weight.

By driving the Lump-BFCI, a single-degree-of-freedom system is achieved. Connecting a four-bar linkage to the Lump-BCFI, in conjunction with the block Lump-ABE, ultimately establishes a six-bar drive mechanism. The connecting rods JK, IJ, FD, BA, and Lump-BFI(C) constitute the Stephenson III-type six-bar drive mechanism, as depicted in Figure 3 and Figure 9. Point K represents the drive shaft axis of the drive system.

The DOF of this planar mechanism is calculated by subtracting the total number of constraints from the total number of DOF of the movable parts, where the number of movable parts N = 7, the number of low pairs PL = 10, and the number of high pairs PH = 0, so the degree of freedom F is:(5)F=3N−2PL−PH=3×7−2×10−0=1

The deformation and driving mechanism of the entire trailing edge is connected to the wing structure via fixed points A and D. The driving mechanism propels Lump-BCFI downward through the rotation of shaft JK. Given that the deformation and driving mechanism operates as a single-DOF system, the movement of shaft JK impels the entire system.

## 4. Detailed Structure Design

### 4.1. Background Aircraft

The CAE-AVM, a long-range business jet, was chosen as the background aircraft, as depicted in Figure 10. The basic performance of the aircraft is shown in Table 2.

Considering the safety requirements for civil aircraft, CAE-AVM chose an airfoil with 30% of the wingspan inward close to the cabin. Additionally, they designed a VCTE to replace traditional slats and flaps. This selection was based on two main factors:The VCTE is smooth and seamless, which can eliminate the steps to reduce noise during the takeoff and landing. The VCTE close to the cabin can maximize efficiency.The VCTE exhibits a smaller stall angle of attack compared to conventional slats and flaps. Opting to position the VCTE at the 30% inboard location, rather than the outer wing, replaces the slats or flaps. This design choice ensures that the inner wing stalls prior to the outer wing during aircraft stall conditions, thus preventing the aircraft from pitching up due to outer wing stall.

### 4.2. Initial, Target Shape and Aerodynamic Loads

The CAE-AVM aircraft has a maximum takeoff weight of 44,000 kg, a maximum cruising speed of 0.9 Ma, a typical cruising speed of 0.85 Ma, and an initial cruising altitude of 13,000 m. The one-quarter chord sweep angle of the three-dimensional wing is 35°, then the calculated M number of the two-dimensional airfoil is 0.85 × cos (35°) = 0.7 Ma, and the Re number is 16E6.

The target shapes at various deflection angles are illustrated in Figure 11. Here, deflection angles (DEs) 0°, 2.26°, 3.7°, and 5° represent the cruise target shape, while 7.5°, 10.0°, 12.5°, and 15° denote the take-off and landing target shape (L&T). The aerodynamic optimization method for VCTE shapes was referenced from [25]. Geometric parameterization of the deformation shape is determined in combination with the features of “finger” deformation structures, and the length and the proportion and rotation angle of the camber line are used for aerodynamic optimization.

The comparison of pressure distribution across the entire airfoil surface at varying angles of downward deflection of VCTE is depicted in Figure 12. It is evident that the deflection angle (DE) correlates with the design lift coefficient (CL). A higher design lift coefficient results in a downward load on the trailing edge, while a lower design lift coefficient leads to an upward unloading of the trailing edge. The peak in pressure distribution observed when the trailing edge is downwardly deflected is primarily attributed to the pressure alteration induced by the abrupt curvature change in the VCTE.

### 4.3. Structural Design Constraints

When designing the mechanism and structure, as shown in Figure 13, several factors must be comprehensively considered, including the driving angle, driving force, and structural interference:Ensure that points A, B, C, and H are positioned as close as possible to the camber line of the airfoil to maintain similarity between the upper and lower telescopic skins.Ensure that the lengths of lines AB, BC, and CH are relatively uniform to facilitate fitting the target shape.Maintain a certain distance between points D, F, G, and I and the upper and lower skins to accommodate the arrangement of rotating shafts and bushings during structural design, thus preventing interference.Maintain adequate spacing between drive points JK and fixed points AD, as well as other components, to prevent interference issues during the deflection process.Preserve appropriate relative rotation angles between the three blocks Lump-ABE, Lump-BFCI, and Lump-CGH to ensure smooth movement and avoid dead-center situations.

The Stephenson six-bar linkage is responsible for actuating the three blocks on the trailing edge. When determining the point locations and ensuring that the six-bar linkage meets the maximum driving angle requirements, efforts should be made to position the upper dead center closer to the initial shape. This is crucial because when the deflection angle of the trailing edge is small, especially in the cruise state, the mechanism is near the dead-center position, leveraging the locking effect to maintain stability, conserving drive energy, and ensuring the mechanism reaches its maximum angle even in the event of drive system failure, thereby ensuring the safety of the VCTE, as depicted in Figure 14.

By analyzing the motion process and driving load and considering factors such as space constraints, driving force, and power, efforts were made to reduce the load values of DF and EG, thus ensuring that the load is primarily transmitted through the three blocks. Adjustments were made to the points to meet the deflection angle requirements of the target shape. The point positions obtained through semi-empirical optimization are presented in Table 3.

The point positions were determined using a combination of automated procedures and manual screening, and the semi-empirical process is depicted in Figure 15. Initially, various motion samples were calculated based on Equation (4) above through a written program, assessing the sensitivity of point positions to both the wing tip deflection angle and the relative deflecting angle of three blocks. Subsequently, a similar approach was applied to the driving mechanism. Utilizing the Latin Hypercube sampling method, a range of point positions were generated within the design space, with optimal samples manually chosen based on structural design constraints and sensitivity analysis outcomes. This method mainly considers complexity and efficiency.

### 4.4. Flexible Skin

According to the movement of the trailing edge mechanism, flexible skin is positioned between the blocks with relative displacement, with metal skin arranged on each block as shown in Figure 16. As all three blocks of the VCTE deflect relative to the beam, flexible skins are installed on the upper and lower skins between the Lump-ABE and the beam, the Lump-BFCI and the Lump-ABE, and the Lump-GHC, and the Lump-BFCI, totaling six flexible skins.

As shown in Figure 17, a flexible silicone rubber (PVMQ, internal grade: GZG0200Q) is positioned in the middle with aluminum alloy strips on both sides. The silicone rubber is bonded to the strips using Cilbond-36 adhesive, with its width adjusting according to the skin length of the trailing edge. The rubber’s thickness is determined based on factors such as adhesive strength and in-plane/out-of-plane stiffness. The cross-sectional shape of the strips is T-shaped, providing both the adhesive surface for the rubber and the overlapping surface for the metal skin at each block, while also enhancing bending stiffness. Parameters are determined based on the skin’s load-carrying capacity.

When designing the initial length of the flexible skin, the length of the upper skin always increases during movement, while the lower skin always compresses. Therefore, when we determine the initial length, in the initial state, the skin is in a compressed state, while the lower skin is in a stretched state. This approach minimizes the maximum stretch and compression rates of the skin.

### 4.5. Detailed Structural Design

The overall structure of the trailing edge adopts a symmetrical design, as illustrated in Figure 18. Ribs, connecting rods, drive rods, and eccentrics all possess a unified plane of symmetry. This design feature serves to eliminate out-of-plane bending moments and avoids designing additional structures to transmit loads. Connecting rods DF and EG bear relatively small loads and are centrally positioned, whereas Lump-ABE and Lump-BFCI, carrying heavier loads, are located on both sides. Supports (point A and D) connect the trailing edge structure to the spar. Detailed parameters such as the diameter of the hole, pin size, and bearing were preliminarily verified.

The primary challenge in structural design is interference, which is addressed as follows: (1) The curved rod DF structure is devised to mitigate interference between rod DF and rotating shafts K during movement; (2) Lightening hole design accounts for interference between Lump-ABE and rotating shaft K during movement; (3) Flexible skin design must consider interference resulting from rivet installation; (4) Given that the main force transmission surfaces between the blocks are collinear, relative interference necessitates consideration. The ultimate structure design is depicted in Figure 19 and Figure 20.

The finite element model was created using Abaqus to analyze various design conditions. The VCTE is made of 7050 aluminum alloy with a tensile strength of 490 MPa. The aerodynamic load is equivalently processed to the skin nodes and transmitted to the wing ribs through the stringers. MPC connection units are established to simulate the bolt connection and constrain the connection position between the ribs and the wing, as shown in Figure 21 and Figure 22. During cruising, the highest stress occurs when the trailing edge is deflected downward by 5°, reaching 322 MPa. The results are shown in Figure 23. According to the airfoil, it should be stretched 2.7 m and arranged with six groups of the above VCTE structures to form a full-size specimen.

### 4.6. Structural Assembly

When manufacturing the trailing edge structure, the assembly begins with the installation of the drive structure onto the frame. First, transmit torque to the drive shaft (point K) via the motor and worm gear reducer, which subsequently pulls the connecting rod IJ using the eccentric disc JK. Next, the wing rib structures Lump-ABE, Lump-BFCI, and Lump-CGH, along with connecting rods DF and EG, are installed and interconnected using pins. Finally, the upper and lower skins are mounted. The assembly process is depicted in Figure 24.

## 5. Ground Test

To ascertain the load-bearing and deflecting capacity of the VCTE structure, we selected two typical scenarios: take-off and landing, and cruising, for verification. During take-off and landing, the maximum deflection angle of the trailing edge design is 15°, whereas in cruising, it is 5°. Employing the load equivalent method, we converted the aerodynamic load to the designated loading point. Table 4 outlines the load conditions at different points in these two states; this paper only considers aerodynamic loads under normal flight conditions and does not consider the gust loads. Given the substantial deformation of the structure/mechanism and the variability in loading angles during the trailing edge deflection process, a tracking–loading method is necessary to ensure loading accuracy.

To accommodate this, the tracking–loading system is required to adjust both load magnitude and angle in real time. For instance, at load point 1, the load is applied via sets of levers using a tape + lever system, routed through pulley steering, and connected to actuator 1. Since the movement at load point 1 is minimal, pulley 1 remains fixed. However, load points 2 and their corresponding levers experience significant directional and positional shifts during deflection. To address this, a tracking–loading mechanism is employed, driving pulley 2 to adjust accordingly, as depicted in Figure 25, Figure 26 and Figure 27. Pulley 2 is affixed to a crossbeam linked to the slider of a screw mechanism. Load point 3, on the other hand, is loaded through actuator 3 via a pad, which can provide tension as well as pressure. Given the substantial positional variation at point 3, a tracking–loading mechanism ensures that actuator 3 maintains the load direction.

Through the tracking–loading system, utilizing the lead screw module paired with a force-controlled actuator in combination with active control, the ground test successfully validated the test piece. The test process employing this scheme demonstrated smooth operation, with the displacement effectively tracked. Along the wing span, the VCTE comprises six identical loading sections. Consequently, during cruising, when the deflecting angle is 5°, the maximum vertical load amounts to 17,072 N. Under this aerodynamic load, the VCTE satisfactorily meets both functional and strength requirements.

## 6. Conclusions

This article explores different methods for implementing Variable Camber Trailing Edge (VCTE) technology and determines that mechanical mechanisms based on kinematics provide the most practical option for designing large civil aircraft. Building on this conclusion, a full-scale trailing edge design is developed and validated.

Coordinating the design of the internal skeleton and skin deformation is crucial for maintaining the structural integrity of the VCTE. Utilizing a “Finger” coordinated design solution provides a straightforward approach.Employing Watt I and Stephenson III mechanism to design a single-degree-of-freedom VCTE offers advantages in terms of simplicity and high load-bearing capacity. Optimizing the linkage points enables better adaptation to supercritical airfoils.To verify the load-bearing capacity of the VCTE structure, a load tracking system based on a “tape + lever” system is designed. This system effectively addresses the challenge of significant changes in load direction.

While the designed structure meets stiffness and strength requirements, further efforts are needed to enhance its performance. Utilizing topology optimization and other methods to reduce structural weight should be prioritized as a crucial area for future research. Furthermore, the present design is a two-dimensional straight segment structure, which impedes quantitative analysis of both aerodynamic and structural benefits. Therefore, developing a three-dimensional wing and carrying out scaled-model flight testing is imperative to address these limitations.

## Figures and Tables

**Figure 1 biomimetics-09-00312-f001:**
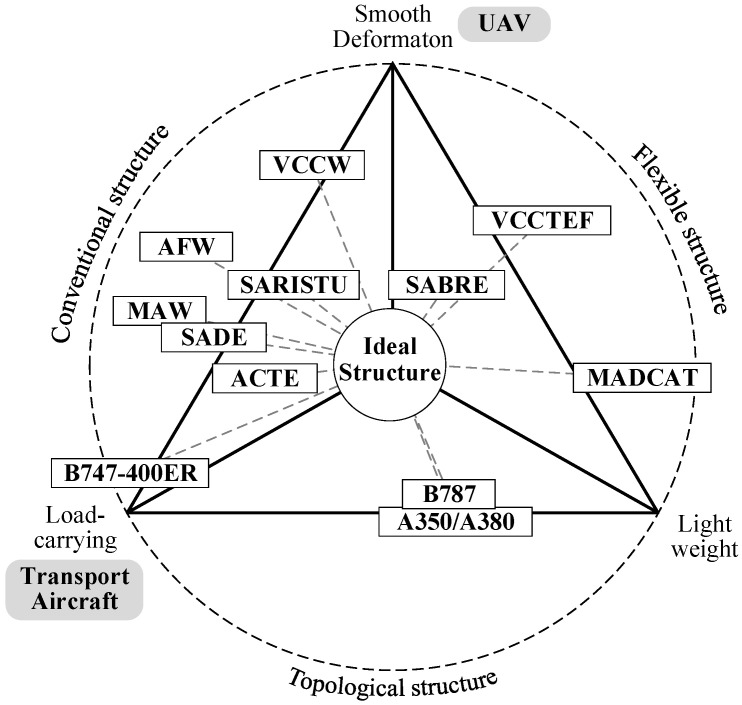
Triangular of smooth deformation, load-carrying, and structural lightweightness.

**Figure 2 biomimetics-09-00312-f002:**
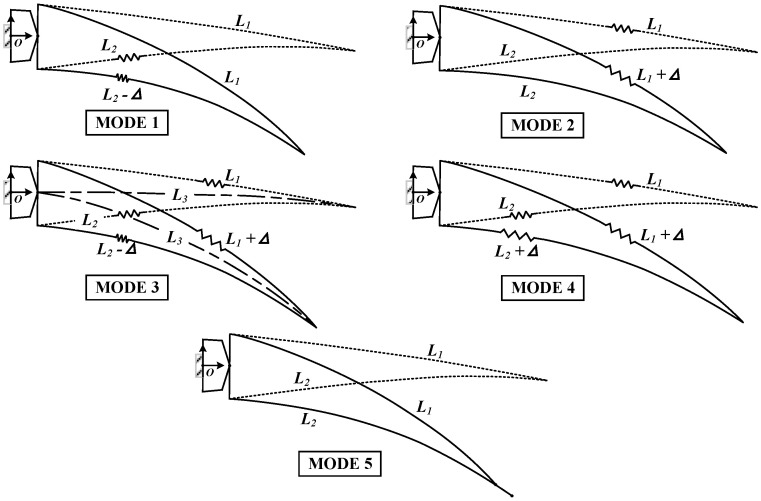
Coordinated design approaches for skin and skeleton.

**Figure 3 biomimetics-09-00312-f003:**
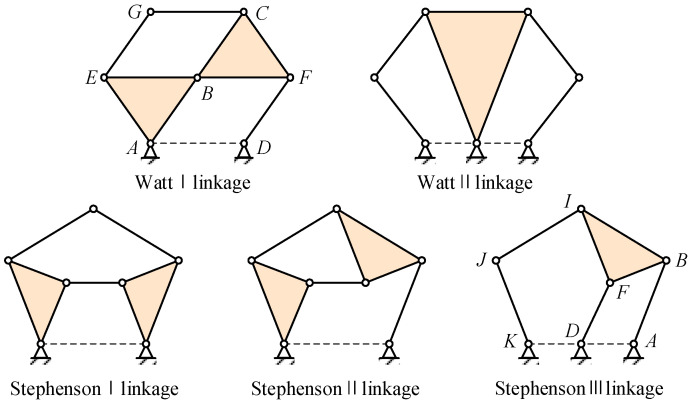
Watt chain and Stephen chain of six-bar linkage.

**Figure 4 biomimetics-09-00312-f004:**
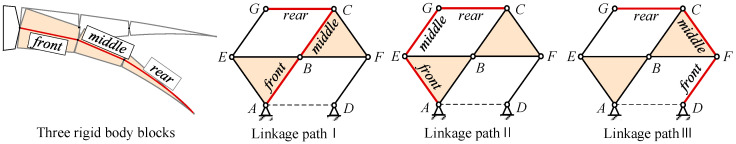
Three optional paths of Watt I.

**Figure 5 biomimetics-09-00312-f005:**
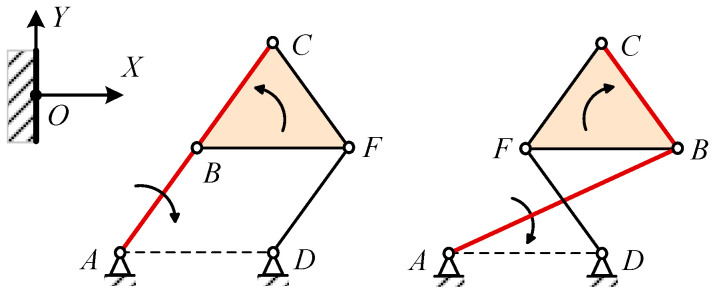
Design paths to ensure consistency in relative rotation directions.

**Figure 6 biomimetics-09-00312-f006:**
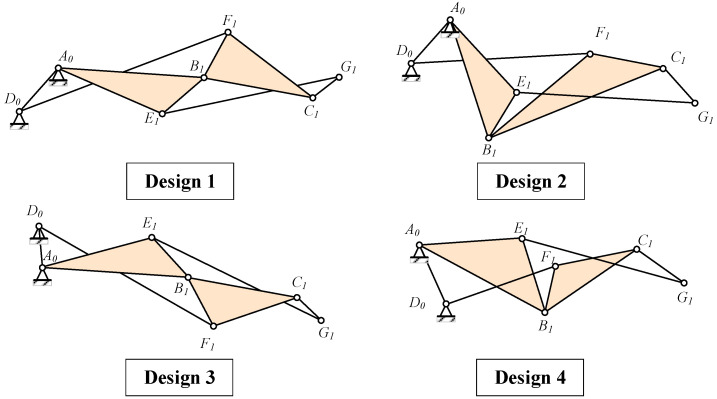
Four primary mechanism forms.

**Figure 7 biomimetics-09-00312-f007:**
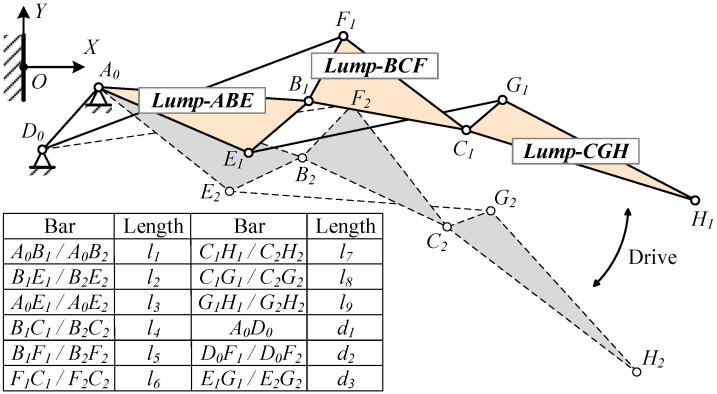
Mechanism form and deformation.

**Figure 8 biomimetics-09-00312-f008:**
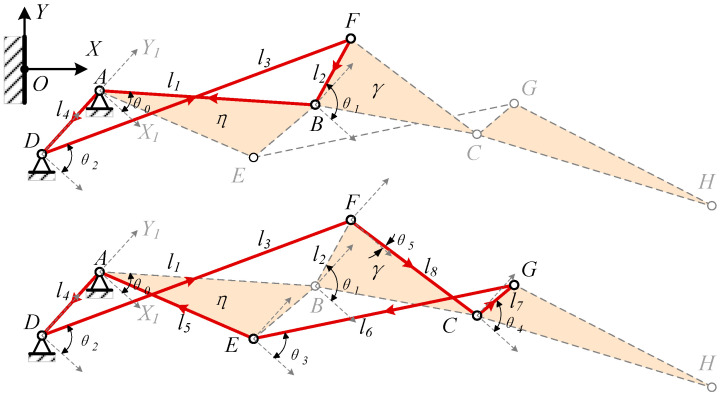
Kinematics Analysis.

**Figure 9 biomimetics-09-00312-f009:**
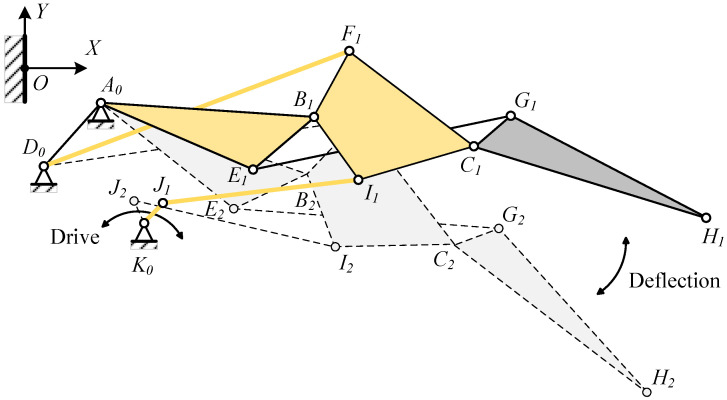
Stephenson six-bar linkage drive mechanism of VCTE.

**Figure 10 biomimetics-09-00312-f010:**
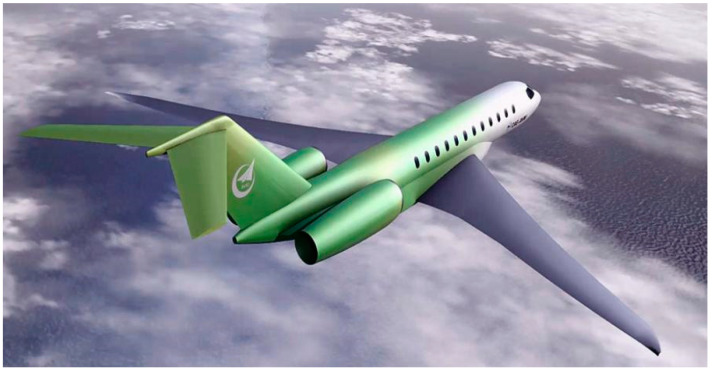
Background aircraft of CAE-AVM.

**Figure 11 biomimetics-09-00312-f011:**
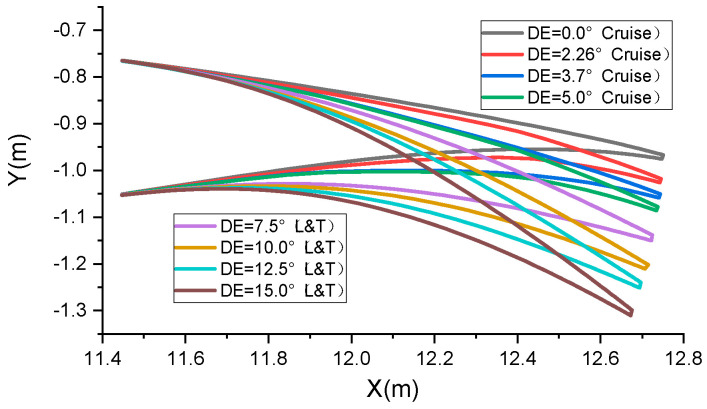
Initial and target shape of airfoil.

**Figure 12 biomimetics-09-00312-f012:**
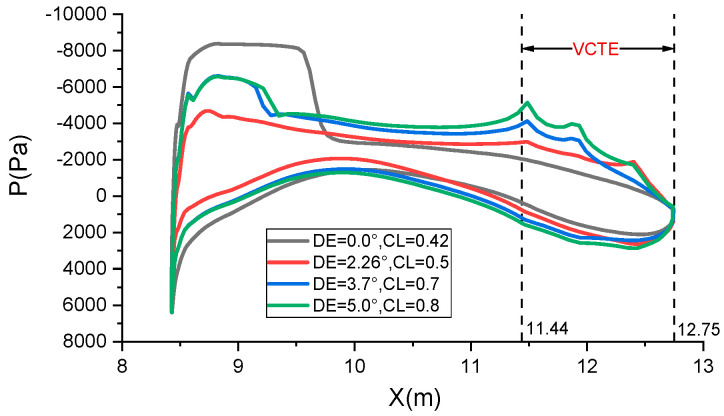
Pressure distribution across the entire airfoil at varying angles.

**Figure 13 biomimetics-09-00312-f013:**
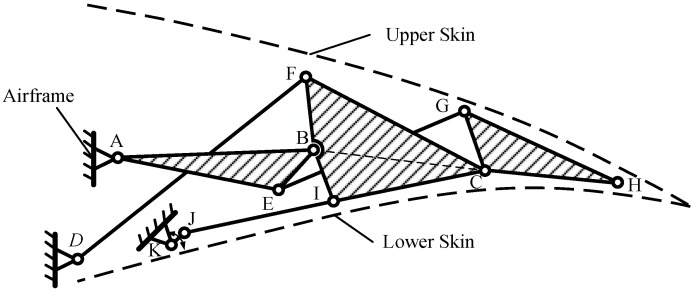
Structural design constraints between skin and mechanism.

**Figure 14 biomimetics-09-00312-f014:**
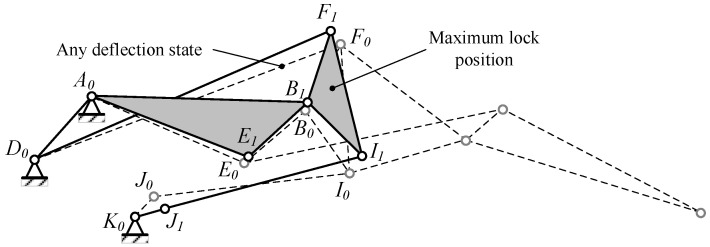
Design for maximum locking position to ensure safety.

**Figure 15 biomimetics-09-00312-f015:**
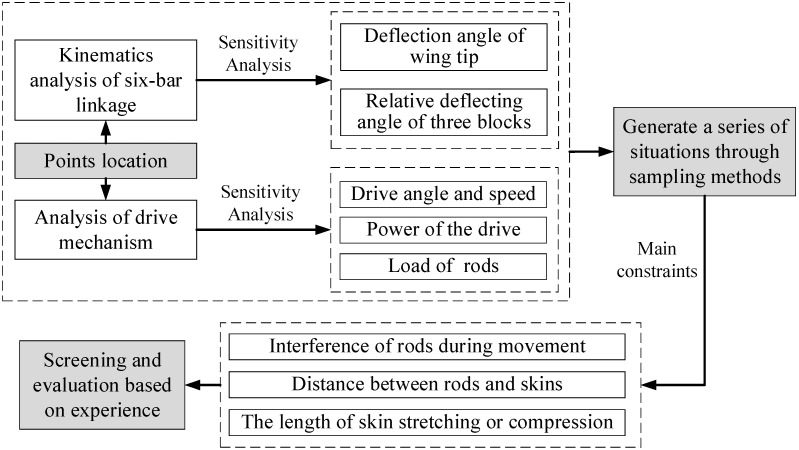
Process of semi-empirical optimization.

**Figure 16 biomimetics-09-00312-f016:**
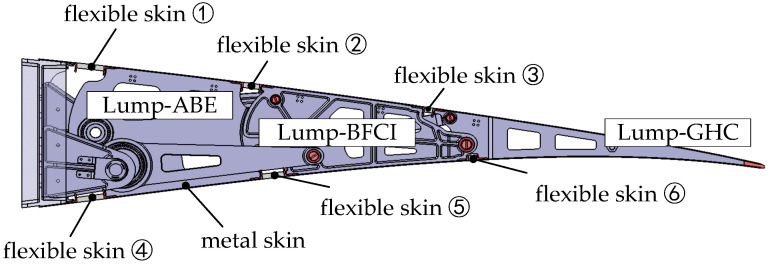
Flexible skin and metal skin of VCTE.

**Figure 17 biomimetics-09-00312-f017:**
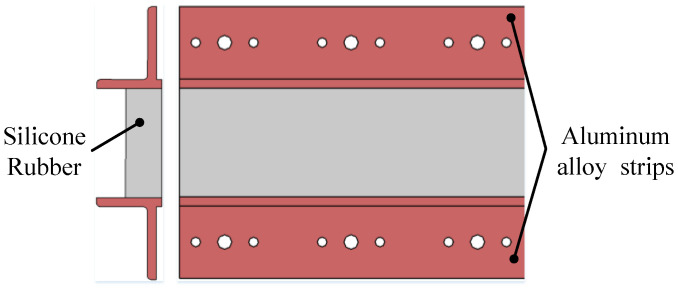
Details of Flexible skin.

**Figure 18 biomimetics-09-00312-f018:**
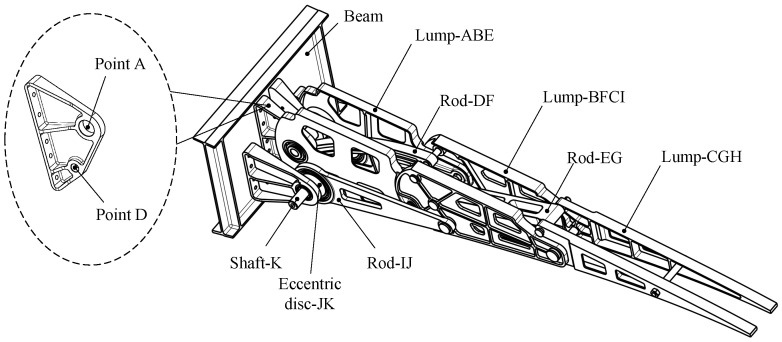
The main layout of the VCTE structure.

**Figure 19 biomimetics-09-00312-f019:**
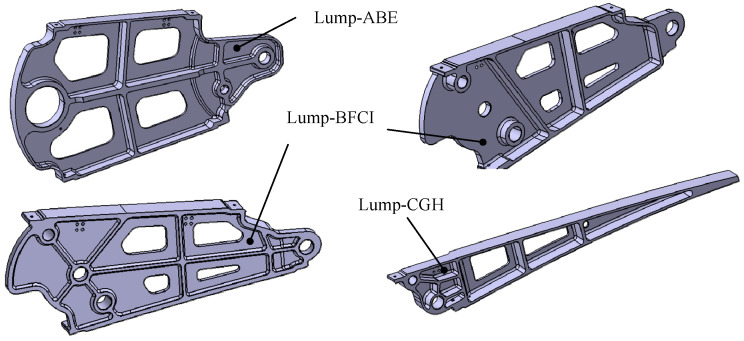
Structure of different blocks.

**Figure 20 biomimetics-09-00312-f020:**
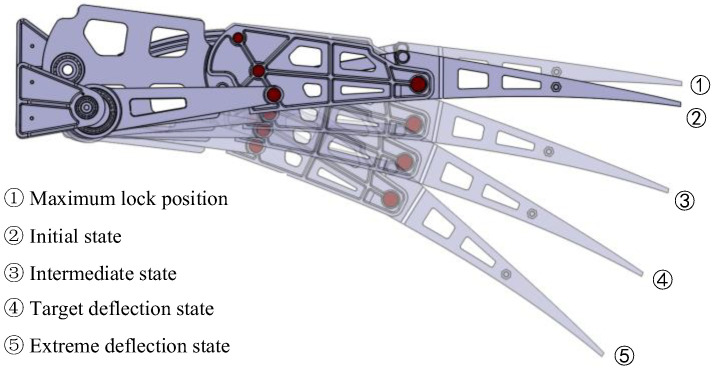
Different upward and downward deflection states.

**Figure 21 biomimetics-09-00312-f021:**
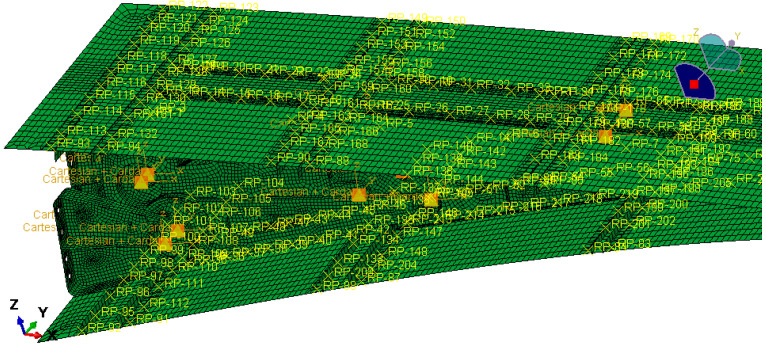
Connection between skins, beams, and ribs.

**Figure 22 biomimetics-09-00312-f022:**
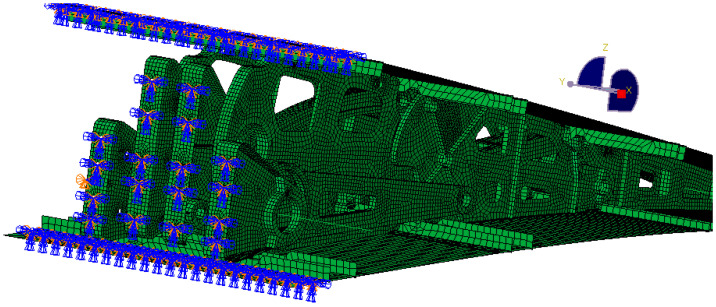
Constraints between the ribs and the wing.

**Figure 23 biomimetics-09-00312-f023:**
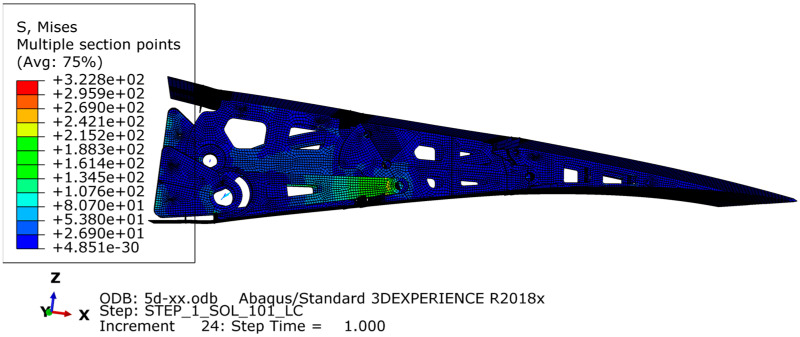
The Finite element model.

**Figure 24 biomimetics-09-00312-f024:**
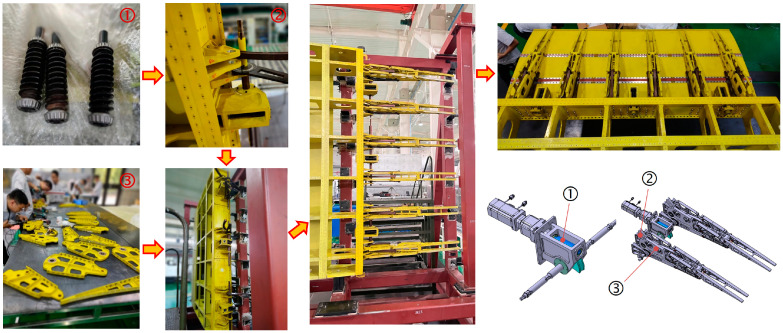
The assembly process of full-size specimen.

**Figure 25 biomimetics-09-00312-f025:**
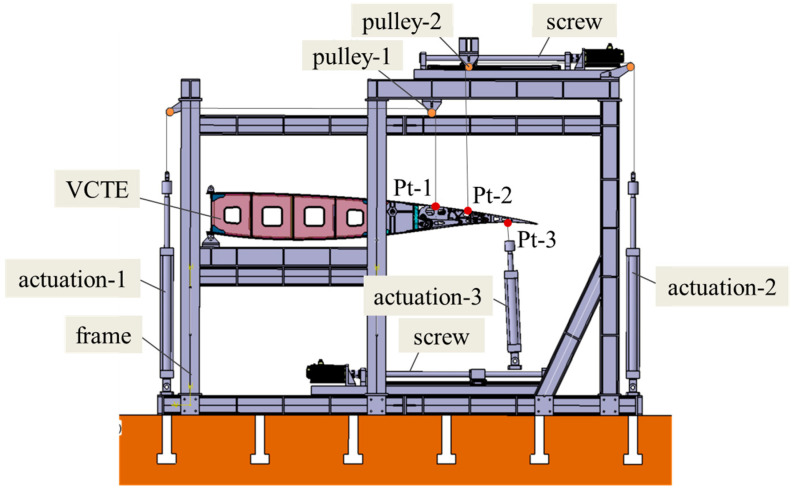
Tracking–loading system of VCTE.

**Figure 26 biomimetics-09-00312-f026:**
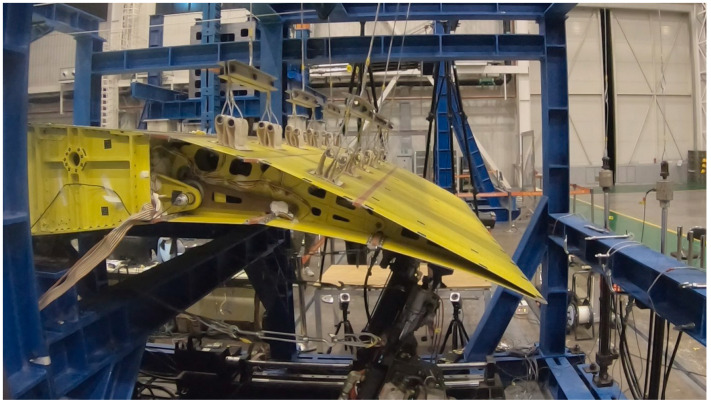
Ground test.

**Figure 27 biomimetics-09-00312-f027:**
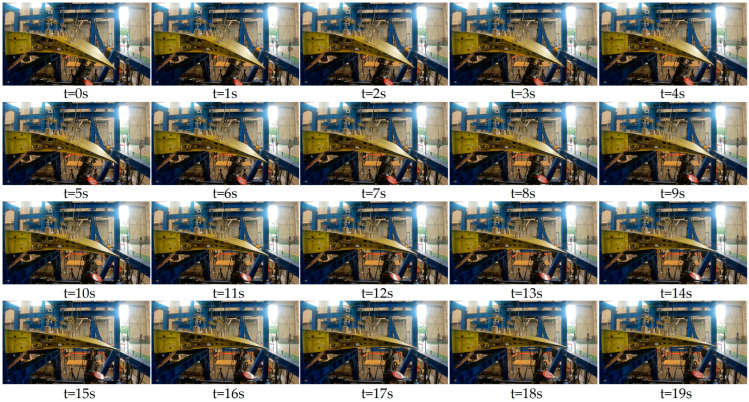
Deformation of VCTE at different times with tracking–loading system.

**Table 1 biomimetics-09-00312-t001:** Comparison of three different implementation approaches.

Implementation	Deflection Angle	Deflection Accuracy	Complexity	Reliability	Load Carrying	Aerodynamic Benefits	Weight	Manufacturing Cost	TRL
Conventional structure	high	high	high	medium	high	medium	high	medium	medium
improved lift augmentation devices	high	high	low	high	high	low	medium	low	high
smart materials	low	low	medium	low	low	high	low	high	low

**Table 2 biomimetics-09-00312-t002:** Basic performance of CAE-AVM.

Parameter	Value	Parameter	Value
Max cruise speed	Ma 0.90	Typical cruise speed	Ma 0.85
Initial cruise altitude	13,000 m	Passengers	8–19 person
Maximum altitude	15,500 m	Maximum takeoff weight	44,000 kg
Maximum range	13,000 km	Commercial load	2200 kg

**Table 3 biomimetics-09-00312-t003:** Point locations of the mechanism.

Coordinate	A	B	C	D	E	F	G	H	I	J	K
X	80.0	450.0	759.88	30.0	380.0	410.0	730.0	1298.86	480.0	129.0	117.0
Y	0.911	1.165	−22.63	−80.0	−35.0	65.0	30.0	−64.35	−45.0	−63.0	−72.0

**Table 4 biomimetics-09-00312-t004:** Load of take-off/landing and cruise Scenarios.

Scenarios	Deflection Angle of VCTE	Load Point	Fz/N	Fx/N
Take-off/landing	15°	Pt-1	594.8	76.6
		Pt-2	336.4	87.9
		Pt-3	259.3	148.9
Cruise	5°	Pt-1	1110.9	79.1
		Pt-2	875.2	86.1
		Pt-3	859.2	165.6

## Data Availability

The original contributions presented in the study are included in the article, further inquiries can be directed to the corresponding author/s.

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
