# Peer review of "Design and Validation of the Trailing Edge of a Variable Camber Wing Based on a Two-Dimensional Airfoil"

_biomimetics, 2024, doi:10.3390/biomimetics9060312_

Round 1

Reviewer 1 Report

Comments and Suggestions for Authors

This paper discusses a variable camber trailing edge (VCTE) design that can be implemented on a full scale aircraft for aerodynamic improvements along with experimental load testing of the VCTE prototype. It is undoubtedly a useful addition to the literature of VCTE concepts that can be helpful for future design. 

Major comments:

1) There should be more discussion to justify the "Biomimetics" aspect of this research in order to qualify publication in the journal.

2) Certain explanations of the design trade-offs can be improved for clarity using figures/videos etc.

Other comments:

1) Introduction - please mention target reynolds number for the design. 

2) Line 70 - "especial" - typo

3) Line 167,172 - Modes2,5 - This needs more clarification as it contradicts with the illogical states mentioned earlier.

4) Line 231 - It will be helpful for the reader if the authors develop the description of the structure such as rear block, front block using a representative figure.

5) Line 237 - please indicate the figure that depicts these connecting blocks.

6) Line 239 - needs more clear explanation of "connecting block intersection method"

7) Line 273 - Please indicate the six bars in the drive mechanism

8) Line 298 - How is point 2 a favorable factor for VCTE design, what is the alleviating factor?

9) Line 302 - how was the target shape decided?

10) Line 339 - "semi empirical optimization" - Is this referring to Eq.4? Please discuss how Eq. 4 is factored into the mechanism design 

11) Line 350 - please provide more information on the adhesive material used.

12) Line 370 - clarify support points A,D w.r.t Figure 13. They are not indicated on the same plane in Fig. 15.

13) Fig. 18 - Can you also show the suction pressure distribution across the VCTE during cruise flight condition.

14) Fig. 20 - Please indicate the loading conditions and values for the various locations (in a table)

15) Structural assembly - a video link of the assembly in operation will be very useful for clarity.

16) Line 398 - needs clear image of the drive actuator installed

17) Line 418 - what does the phrase "follow up loading mechanism" mean. Is it referring to the mechanism that adjusts the base of the actuator? If so, a more clear term could be used.

18) Fig. 23 - authors can show a series of images with the various actuators and loading points.

19) Line 426 - if video is available it will be very useful to include as link. 

Comments on the Quality of English Language

Minor edits required.

Author Response

Dear editor:

Thank you very much to spare time for the manuscript. We feel great thanks for your professional review work and nice suggestions. Revised portion are marked with yellow background. The main corrections and responses to the reviewer’s comments are as flowing:

  1. Major comment: There should be more discussion to justify the "Biomimetics" aspect of this research in order to qualify publication in the journal.

Response: We have modified the content of the introduction to better fit the "Biomimetics" aspect. (Line 27)

  1. Major comment: Certain explanations of the design trade-offs can be improved for clarity using figures/videos etc.

Response: We have added some description of relevant data to the Introduction. The benefits of VCTE have been described in our team's related research. According to our evaluation, employing the VCTE can enhance the lift-to-drag ratio of the aircraft airfoil by 7% and the overall lift-to-drag ratio by 3.5%, and totally save 8760 tons of fuel during the service life per aircraft. The relevant reference (in Chinese) is as follows: (Line 81)

Li, Shitu, et al. “Development and application prospect of variable leading and trailing edge structure technology.” Aeronautical Science & Technology 33.12 (2022): 31-40.

  1. Comment: Introduction - please mention target Reynolds number for the design.

Response: The Reynolds is 16E6. We have added information about the background aircraft in Section 4.2. (Line 325)

  1. Comment: Line 70 - "especial" – typo.

Response: Corrected. Sorry for careless mistake. (Line 76)

  1. Comment: Line 167,172 - Modes2,5 - This needs more clarification as it contradicts with the illogical states mentioned earlier.

Response: We have revised the wording to provide clearer and more accurate descriptions of four illogical states to avoid ambiguity. (Line 171)

  1. Comment: Line 231 - It will be helpful for the reader if the authors develop the description of the structure such as rear block, front block using a representative figure.

Response: We have modified Figure 4 by adding an image corresponding to three rigid body blocks and adding labels for the front, middle, and back blocks to enhance readability. (Line 237)

  1. Comment: Line 237 - please indicate the figure that depicts these connecting blocks.

Response: We have modified Figure 4 and described three connection blocks on it. (Line 237)

  1. Comment: Line 239 - needs more clear explanation of "connecting block intersection method".

Response: We have added a section to explain the "connecting block intersection" method in detail to make it easier for readers to understand. The content is as follows:

When the VCTE deflects downward, it's essential to ensure that the deflecting direction of the middle block relative to the front block and the rear block relative to the middle block is consistent. Consequently, lines AB and DF must intersect to ensure uniform deflection of AB and BC in the same direction, as depicted on the right side of Figure 5. Conversely, on the left side of Figure 5, the deflecting direction of AB and BC are opposite. (Line 248)

  1. Comment: Line 273 - Please indicate the six bars in the drive mechanism.

Response: We have added a section to indicate the specific name of the six bars.

The connecting rods JK, IJ, FD, BA and Lump- BFI(C) constitute the Stephenson III type six-bar drive mechanism, as depicted in Figure 3 and Figure 9. (Line 288)

  1. Comment: Line 298 - How is point 2 a favorable factor for VCTE design, what is the alleviating factor?

Response: The point 2 explains why the location 30% closer to the cabin was chosen instead of the location outside. Using VCTE at this position can make the inside of the wing of the aircraft stall first when the aircraft stalls, thus enhancing the safety of the aircraft. More precisely, this is the reason for choosing this location for the variable camber wing design, rather than the advantages of the variable camber wing itself.

We have rewritten the point 2 to make it clearer and easier to understand. (Line 315)

  1. Comment: Line 302 - how was the target shape decided?

Response: We have added corresponding reference to this paragraph, as follows: (Line 329)

Chunpeng, Li, et al. “Trailing edge deformation matrix aerodynamic design for log-range civil aircraft variable camber wing.” Acta Aeronautica et Astronautica Sinica, 44.07(2023): 92-107.

  1. Comment: Line 339 - "semi empirical optimization" - Is this referring to Eq.4? Please discuss how Eq. 4 is factored into the mechanism design.

Response: We added a detailed explanation and Figure 15 to describe how to determine the position of the point. Eq.4 is mainly used to calculate the position of each point during motion. We wrote a program for calculation. (Line 385)

  1. Comment: Line 350 - please provide more information on the adhesive material used.

Response: We have added descriptions of silicone rubber and adhesive grade. The flexible silicone rubber (PVMQ, internal grade: GZG0200Q), the adhesive material is Cilbond-36. (Line 396)

  1. Comment: Line 370 - clarify support points A,D w.r.t Figure 13. They are not indicated on the same plane in Fig. 15.

Response: We have added an auxiliary view to the Fig 18 to describe points A and D more clearly. (Line 420)

  1. Comment: 18 - Can you also show the suction pressure distribution across the VCTE during cruise flight condition.

Response: We have added Figure 12, which describes the pressure distribution at different deflection angles in cruise state and take-off and landing state. (Line 339)

  1. Comment: 20 - Please indicate the loading conditions and values for the various locations (in a table).

Response: We have added Table 3, supplemented the loading data for takeoff/landing and cruise scenarios, and corrected the description of the loading method to make it more accurate to use load tracking. (Line 483)

  1. Comment: Structural assembly - a video link of the assembly in operation will be very useful for clarity.

Response: We are very sorry that we did not produce a complete assembly video during the assembly process.

  1. Comment: Line 398 - needs clear image of the drive actuator installed

Response: We have updated the assembly process (Fig 24) and added a picture of the drive system. (Line 458)

  1. Comment: Line 418 - what does the phrase "follow up loading mechanism" mean. Is it referring to the mechanism that adjusts the base of the actuator? If so, a more clear term could be used.

Response: We corrected the term to use " tracking-loading" instead of "follow-up loading " to make it clearer. (Line 488)

  1. Comment: 23 - authors can show a series of images with the various actuators and loading points.

Response: Thanks for the nice advice! We added the deformation of the trailing edge at different times as shown in Fig 27. (Line 486)

  1. Comment: Line 426 - if video is available it will be very useful to include as link.

Response: Sorry, we don't have a video link, but have added a series of test images as shown in Fig 27. (Line 486)

Thank you very much for your time and consideration.

Sincerely yours,

Jin Zhou

Reviewer 2 Report

Comments and Suggestions for Authors

 The following information should be removed from the abstract of the manuscript. If it is necessary please include it in the introduction section; “Variable camber wing technology holds the potential to enhance the cruise lift-to-drag 8 ratio, reduce takeoff and landing noise, and improve fuel economy, which is an important trend for 9 the development of green long-range civil aircraft in the future.”

Please rewrite the abstract of the manuscript. First of all, the main purpose of the study should be summarized. In other words, the focus of the study and why it is important should be explained. Along with the materials and methods used, which parameters were analyzed and the working range limits and restrictions of those parameters considered should be explained. The most important main findings and conclusions of the study should be summarized.

The conclusion of the manuscript is also needed to be improved. For example, the conclusion section is of great importance as it summarizes the main findings and results of the study. It is imperative to summarize the study's contributions to the literature. This section should describe the impact of the study on existing literature and its potential implications for practical applications. Additionally, making recommendations based on research findings or assessing future research directions may be helpful.

In wing design, it is commonly advised to establish the aerodynamic profile as the initial step, followed by configuring the internal structure of the wing to align with aerodynamic requirements. While this approach holds merit, the manuscript lacks sufficient detail regarding the aerodynamic parameters relevant to the proposed mechanical design approach.

The performance of the wing is very important in many aspects in aircraft design. Therefore, it is vital to align your design goals with the aerodynamics of the flexible wing.

In general, winglets help reduce induced drag, which is caused by the difference in air pressure between the upper and lower surfaces of the wings. By altering the airflow around the wingtips, winglets decrease drag, which can improve the wing performance in many aspects.  Winglets may also contribute to increased lift, particularly during certain flight conditions such as during takeoff, landing, or maneuvers. This can improve overall aircraft performance. So in your design, you have no winglet, Why is that? Is it because the size of the wing is small? These days this winglet is also used for unmanned aircraft.

In the conclusion section, the statement "The results demonstrate that the stiffness and strength of the structure meet the requirements under maximum aerodynamic load conditions (cruise condition)" appears inaccurate. The maximum aerodynamic load typically occurs during takeoff conditions, rather than cruise conditions."During takeoff, the aircraft accelerates and lifts off the ground, requiring significant aerodynamic forces to overcome gravity and generate lift.

Comments on the Quality of English Language

Minor editing in the English language is required.

Author Response

Dear editor:

Thank you very much to spare time for the manuscript. We feel great thanks for your professional review work and nice suggestions. Revised portion are marked with yellow background. The main corrections and responses to the reviewer’s comments are as flowing:

  • Comment: The following information should be removed from the abstract of the manuscript. If it is necessary please include it in the introduction section; “Variable camber wing technology holds the potential to enhance the cruise lift-to-drag 8 ratio, reduce takeoff and landing noise, and improve fuel economy, which is an important trend for 9 the development of green long-range civil aircraft in the future.”

Please rewrite the abstract of the manuscript. First of all, the main purpose of the study should be summarized. In other words, the focus of the study and why it is important should be explained. Along with the materials and methods used, which parameters were analyzed and the working range limits and restrictions of those parameters considered should be explained. The most important main findings and conclusions of the study should be summarized.

The conclusion of the manuscript is also needed to be improved. For example, the conclusion section is of great importance as it summarizes the main findings and results of the study. It is imperative to summarize the study's contributions to the literature. This section should describe the impact of the study on existing literature and its potential implications for practical applications. Additionally, making recommendations based on research findings or assessing future research directions may be helpful.

Response: Many thanks for such detailed suggestions, we have rewritten the abstract and conclusion sections.

In abstract section, we posit that the pivotal finding of this study underscores the efficacy of employing the single-degree-of-freedom "Finger" concept as the mechanism design for the trailing edge of large civil aircraft. Subsequent ground tests further corroborate its ability to accommodate substantial deflection angles and meet atmospheric dynamic load requirements. Additionally, a load tracking test method for ground testing is proposed. (Line 8)

In conclusion section, we encapsulate the crucial design methodologies for variable camber wings and delineate key design considerations, offering valuable insights for future research endeavors. Moreover, we highlight the deficiency in describing structural weight and comprehensive income assessment in the paper, advocating for subsequent verification through scaled model flight testing. (Line 496)

  • Comment: In wing design, it is commonly advised to establish the aerodynamic profile as the initial step, followed by configuring the internal structure of the wing to align with aerodynamic requirements. While this approach holds merit, the manuscript lacks sufficient detail regarding the aerodynamic parameters relevant to the proposed mechanical design approach. The performance of the wing is very important in many aspects in aircraft design. Therefore, it is vital to align your design goals with the aerodynamics of the flexible wing.

Response: We have added the wing shapes at different deflection angles in the cruise state and take-off and landing states, as well as the pressure distribution at different deflection angles in the cruise state in Figure 11 and Figure 12. (Line 330)

Currently, when designing the wings of large civil aircraft, maintaining optimal aerodynamic performance across the entire flight envelope under multiple configurations is usually not considered. In fact, when we originally designed the target deformation, we only considered the optimal aerodynamic performance. However, it is difficult to design a reasonable internal structure based on these shapes. Therefore, we believe that when designing the shape, we must consider the structural constraints and jointly determine the aerodynamics through repeated iterations. Appearance and internal structure.

  • Comment: In general, winglets help reduce induced drag, which is caused by the difference in air pressure between the upper and lower surfaces of the wings. By altering the airflow around the wingtips, winglets decrease drag, which can improve the wing performance in many aspects. Winglets may also contribute to increased lift, particularly during certain flight conditions such as during takeoff, landing, or maneuvers. This can improve overall aircraft performance. So in your design, you have no winglet, Why is that? Is it because the size of the wing is small? These days this winglet is also used for unmanned aircraft.

Response: Winglets are also a very promising morphing wing technology. They can not only increase lift and reduce drag, but also add additional maneuverability. At the same time, they can also meet the adaptability issues of 4C airports for large civil aircraft. We are also currently focusing on research on winglets, hoping to eventually integrate them into full-size wings. However, because this article mainly studies the design method of variable camber wings, this part is not covered.

  • Comment: In the conclusion section, the statement "The results demonstrate that the stiffness and strength of the structure meet the requirements under maximum aerodynamic load conditions (cruise condition)" appears inaccurate. The maximum aerodynamic load typically occurs during takeoff conditions, rather than cruise conditions." During takeoff, the aircraft accelerates and lifts off the ground, requiring significant aerodynamic forces to overcome gravity and generate lift.

Response: During actual flight phases, the most substantial loads typically occur during takeoff and landing. However, due to the variable camber trailing edge discussed in this article, the maximum deflection angle is limited to 15°, which is less than the typical range of 30°-35°. Consequently, the loads experienced during takeoff and landing are not as significant as those encountered under normal circumstances. In contrast, the maximum deflection angle during the cruise stage is 5°, aligning with typical operating conditions. Therefore, at a speed of 0.85 Ma, the maximum load during the cruise stage surpasses that of takeoff and landing, representing the peak load in this study. We give the load comparison at the maximum deflection angle in cruise and take-off and landing states in Table 3. (Line 482)

Thank you very much for your time and consideration.

Sincerely yours,

Jin Zhou

Reviewer 3 Report

Comments and Suggestions for Authors

This research presents an extension of the design and validation of the trailing edge from the work [15], which is not clearly extension.  The work proposes an interesting idea for morphing wing with camber change, but it is lag in clarity regarding the conceptual design starting from design problem, design details, validation, and experiments. This lags cause unrepeatable experiment by other researches. If the main contribution of this paper has been published with very well document, the present paper must be clearer of extensions. I cannot recommend this paper to be published in this journal due to the main concerns as previous mentions. The following suggestions can make this paper clearer.

1)     In abstract, it must be clear explanation of aerodynamic load and its calculation in the experiment part.

2)     In literature, the paper [15] must be clear the relation with the present research. The extensions should be presented point by point.

3)     If the eight-bar linkage is not ground based of this study, it should be discarded from section 3.2.

4)     The word “loop” is properly used rather than “four-bar ring” in section 3.2.

5)     The background aircraft part, what is the main MTOW, Mach etc. related to your aerodynamic load calculation. It must be clear explanation in aerodynamic load analysis in section 4.5 and ground test.

6)     The semi-empirical optimization needs a detail in form of flow diagram/pseudo code, initial testing data, algorithm, software, iteration number, computer performance etc.  

7)     First paragraph in Section 4.4 needs a figure (fig 15) to make it clearer.

8)     The apply load, boundary condition and material properties are necessary for the part of MSC Patran solving.

9)     Ground test, the loading points are not clear point in the figure. The loading is seemed not accord with aerodynamic shape and general static test. How much g testing in this case?

10) It seems the aims, results and conclusion cannot convince the reader to accept the benefit of this research.

Comments on the Quality of English Language

Moderate English quality. 

Author Response

Dear editor:

Thank you very much to spare time for the manuscript. We feel great thanks for your professional review work and nice suggestions. Revised portion are marked with yellow background. The main corrections and responses to the reviewer’s comments are as flowing:

  1. Comment: In abstract, it must be clear explanation of aerodynamic load and its calculation in the experiment part.

Response: We have added Figure 12, which describes the pressure distribution at different deflection angles in cruise state and take-off and landing state, and gives some information about the background aircraft in Section 4.2. (Line 339)

  1. Comment: In literature, the paper [15] must be clear the relation with the present research. The extensions should be presented point by point.

Response: Reference [15] is an overview of the entire research project. This article mainly introduces the design and verification method of the variable camber trailing edge structure, which is a detailed expansion of its implementation details and research content. We have added more aerodynamic data and structural analysis results to the article to describe the specific details.

  1. Comment: If the eight-bar linkage is not ground based of this study, it should be discarded from section 3.2.

Response: we simplify the description of the eight-bar linkage in this section to focus more on the six-link mechanism. (Line 223)

  1. Comment: The word “loop” is properly used rather than “four-bar ring” in section 3.2.

Response: Thank you very much, we have changed the “ring” to “loop”. (Line 216)

  1. Comment: The background aircraft part, what is the main MTOW, Mach etc. related to your aerodynamic load calculation. It must be clear explanation in aerodynamic load analysis in section 4.5 and ground test.

Response: We have added information about the background aircraft such as MTOW, Reynolds, Mach in Section 4.2. We added Table 3 to describe the load conditions of each loading point in Section 5.

  1. Comment: The semi-empirical optimization needs a detail in form of flow diagram/pseudo code, initial testing data, algorithm, software, iteration number, computer performance etc.

Response: We have added Fig 15 and some explanations to describe the process of semi-empirical optimization. (Line 385)

  1. Comment: First paragraph in Section 4.4 needs a figure (fig 15) to make it clearer.

Response: We have added Fig 16 to describe flexible skin and metal skin of VCTE. (Line 394)

  1. Comment: The apply load, boundary condition and material properties are necessary for the part of MSC Patran solving.

Response: We have added Fig 21 and Fig 22 and some explanations to describe the analysis of Abaqus. Sorry, we made a silly mistake here, we are actually using Abaqus, not MSC Patran. (Line 443)

  1. Comment: Ground test, the loading points are not clear point in the figure. The loading is seemed not accord with aerodynamic shape and general static test. How much g testing in this case?

Response: The load we use is the aerodynamic load when deflected by 5° in the cruising state. The specific values of the loading point load are shown in Table 3. (Line 482)

  1. Comment: It seems the aims, results and conclusion cannot convince the reader to accept the benefit of this research.

Response: we have rewritten the abstract and conclusion sections. In conclusion section, we encapsulate the crucial design methodologies for variable camber wings and delineate key design considerations, offering valuable insights for future research endeavors. Moreover, we highlight the deficiency in describing structural weight and comprehensive income assessment in the paper, advocating for subsequent verification through scaled model flight testing. (Line 496)

Thank you very much for your time and consideration.

Sincerely yours,

Jin Zhou

Reviewer 4 Report

Comments and Suggestions for Authors

·         Though the mechanism was experimentally tested, the amount of deviation of parameters such as Drag, Takeoff distance, fuel efficiency,  with conventional wing was not quantified and hence the success ratio is not known. Please clarify. May be few discussion points can be added in the manuscript

·         The proposed mechanical model can be structurally accepted but the flow analysis can also be done for getting more clarity on structural rigidity. May be few discussion points can be added in the manuscript

·         Whether this mechanism will be suitable for some symmetrical airfoil configurations such as Diamond shaped airfoil? Please clarify. May be few points can be added in the manuscript

·         Whether the mechanism withstand high speed flow conditions? How to make the linkage and skin rigid during high-speed flow? May be few discussion points can be added in the manuscript

·         Whether the overlapping of metal skin with silicone rubber will create drag at the overlapping areas and result in structural failure during actual flight? May be few discussion points can be added in the manuscript

·         The structural performance was seemed to be good but incorporation of separate linkages and circuits may increase the structural weight and weight comparison was not justified.

·         In most of the aircraft the fuel tank is incorporated inside the wing cabinet. Changing the volume of the wing during deformation may also increases the pressure on fuel. This problem has not considered in this article

·         The engine of the aircraft is attached to the bottom side of the wing in most conventional aircrafts. Varying the wing length may also affect the linkage with engine and hence this has to be considered.

Conclusion should be rewritten with salient points of inferences / outputs / significant contributions

Comments on the Quality of English Language

Needs to improve the formulation of sentences and ensure the grammar again throughout the article

Author Response

Dear editor:

Thank you very much to spare time for the manuscript. We feel great thanks for your professional review work and nice suggestions. Revised portion are marked with yellow background. The main corrections and responses to the reviewer’s comments are as flowing:

  1. Comment: Though the mechanism was experimentally tested, the amount of deviation of parameters such as Drag, Takeoff distance, fuel efficiency, with conventional wing was not quantified and hence the success ratio is not known. Please clarify. May be few discussion points can be added in the manuscript.

Response: We have added some description of relevant data. The benefits of VCTE have been described in our team's related research. According to our evaluation, employing the VCTE can enhance the lift-to-drag ratio of the aircraft airfoil by 7% and the overall lift-to-drag ratio by 3.5%, and totally save 8760 tons of fuel during the service life per aircraft. The relevant reference (in Chinese) is as follows: (Line 81)

Li, Shitu, et al. “Development and application prospect of variable leading and trailing edge structure technology.” Aeronautical Science & Technology 33.12 (2022): 31-40.

  1. Comment: The proposed mechanical model can be structurally accepted but the flow analysis can also be done for getting more clarity on structural rigidity. May be few discussion points can be added in the manuscript.

Response: We have added Fig 12, which describes the pressure distribution at different deflection angles in cruise state and take-off and landing state. (Line 339)

  1. Comment: Whether this mechanism will be suitable for some symmetrical airfoil configurations such as Diamond shaped airfoil? Please clarify. May be few points can be added in the manuscript.

Response: For different airfoil configurations, it could be designed as different type of mechanism, as described in Fig 6. For Diamond shaped airfoil, design 2 or design 4 might be better. We have added some discussion in the manuscript. (Line 259)

  1. Comment: Whether the mechanism withstand high speed flow conditions? How to make the linkage and skin rigid during high-speed flow? May be few discussion points can be added in the manuscript.

Response: The mechanism of VCTE in the article has been verified under cruise aerodynamic load and can meet the strength requirements. We have added Fig 21 and Fig 22 and some explanations to describe the FEM analysis. (Line 443)

  1. Comment: Whether the overlapping of metal skin with silicone rubber will create drag at the overlapping areas and result in structural failure during actual flight? May be few discussion points can be added in the manuscript.

Response: According to the aerodynamic analysis results, the overlap between the metal skin and silicone rubber significantly alters the pressure distribution of VCTE, as illustrated in Fig 12. In our design approach, we aimed to minimize the length of the flexible skin while employing prestressing during assembly to ensure compliance with requirements.

  1. Comment: The structural performance was seemed to be good but incorporation of separate linkages and circuits may increase the structural weight and weight comparison was not justified.

Response: The weight of the structure is indeed one of the weaknesses of this article. We also mentioned in the conclusion that topological design and other methods should be used to reduce the weight of the structure. Because there is no benchmark aircraft weight, it is currently difficult for us to calculate benefits by comparing weight.

  1. Comment: In most of the aircraft the fuel tank is incorporated inside the wing cabinet. Changing the volume of the wing during deformation may also increases the pressure on fuel. This problem has not considered in this article.

Response: Thank you for your very professional advice. In fact, we have already considered this issue when designing VCTE. The motor, reducer, and drive structure should be installed within the trailing edge as much as possible to avoid interference with the wing fuel tank within the wing. We have added relevant descriptions in Section 3.5. (Line 281)

  1. Comment: The engine of the aircraft is attached to the bottom side of the wing in most conventional aircrafts. Varying the wing length may also affect the linkage with engine and hence this has to be considered.

Response: Consider the safety requirements for civil aircraft, we have chosen an airfoil with 30% of the wingspan inward close to the cabin to design the VCTE. This position is a certain safe distance from the engine.

  1. Comment: Conclusion should be rewritten with salient points of inferences / outputs / significant contributions.

Response: we have rewritten the abstract and conclusion sections. In conclusion section, we encapsulate the crucial design methodologies for variable camber wings and delineate key design considerations, offering valuable insights for future research endeavors. Moreover, we highlight the deficiency in describing structural weight and comprehensive income assessment in the paper, advocating for subsequent verification through scaled model flight testing. (Line 496)

Thank you very much for your time and consideration.

Sincerely yours,

Jin Zhou

Round 2

Reviewer 2 Report

Comments and Suggestions for Authors

The manuscript has been revised taking the comments of the referees into account. Now, the manuscript in its current form has been deemed suitable for publication in your journal.

Author Response

We are very grateful for your professional review work and previous good suggestions.

Reviewer 3 Report

Comments and Suggestions for Authors

The work proposes an interesting idea in a trailing edge shape changing design using six-bar linkage and its improvement version is really changed, but it is still lag in design and experiment details. I recommend the current situation is major revision. The following suggestions can make this paper clearer.

1)     In abstract, the abbreviation must be explained, if it not necessary must be discarded.

2)     The design problem, validation, design details, and experiments must be introduced enough necessary information to make it repeatable experiment.

3)     “H” in figure 7 must be cleared its definition.

4)     “K” must be clear explanation at the first used.

5)     The solution technique of Watt I six-bar linkage needs validation.

6)     The specification of CAE-AVM can detail in form of tabular rather than explanation to make it clearer.

7)     The airfoil profile type of the CAE-AVM requires details to understand aerodynamic characteristics.

8)     The aerodynamic optimization in section 4.2 needs a briefly detail in analysis.

9)     The figure 11 and 12 need a citation if it is not a result of this study.

10) Please check “the mechanism t near”.

11) The semi-empirical optimization has been detailed in form of flow diagram, but its initial testing data, algorithm, software, iteration number, computer performance etc are not revealed.  

12) “490 Mpa” must be changed to “490 MPa”.

13) The aerodynamic applied load, FEA details, and boundary condition are necessary for the part of Abaqus solving. What kind of approximation stress that author used in the design criteria of VCTE?

14) I concern the VCTE failure when its deflection with higher angle at take-off and landing condition.

15) The ground test must be addressed. The ground test is not clear point of goal. The loading is seemed not accord with aerodynamic shape and general static test. How much g testing in this case?

Comments on the Quality of English Language

Moderate editing of English language required
